# $A^2$-DP: Annotation-Aware Data Pruning for Object Detection

## Abstract

As the size of datasets for training deep neural networks expands, data pruning has become an intriguing area of research due to its ability to achieve lossless performance with a reduced overall data volume. However, traditional data pruning usually demands complete dataset annotations, incurring high costs. To tackle this, we propose an innovative **A**nnotation-**A**ware **D**ata **P**runing paradigm tailored for object detection, dubbed as $A^2$-DP, which aims to reduce the burdens of both annotation and storage. Our approach, consisting of two phases, integrates a hard sample mining module to extract crucial hidden objects, a class balance module to identify important objects in rare or challenging classes and a global similarity removal module that enhances the elimination of redundant information through object-level similarity assessments. Extensive experiments on 2D and 3D detection tasks validate the effectiveness of the $A^2$-DP, consistently achieving a minimum pruning rate of 20% across various datasets, showcasing the practical value and efficiency of our methods.

## 1 Introduction

Deep learning has achieved remarkable success across a wide range of domains, including computer vision, natural language processing, and generative modeling (Ren et al., 2015; He et al., 2016; Dosovitskiy et al., 2020; Mildenhall et al., 2021; Devlin, 2018; Ho et al., 2020; Achiam et al., 2023; Kirillov et al., 2023). However, training these models typically demands vast amounts of data (Achiam et al., 2023; Kirillov et al., 2023). While large datasets can be advantageous, they often contain redundant or less informative samples, leading to inefficiencies in data storage and model training, which can become costly. To mitigate these challenges, techniques such as dataset pruning (Sorscher et al., 2022; Ayed & Hayou, 2023; Qin et al., 2023; He et al., 2024) have been introduced. Dataset pruning aims to remove redundant or easy samples, thereby conserving storage and computational resources without compromising model performance.

Recent advancements in dataset pruning (Sorscher et al., 2022) have primarily focused on simple classification tasks, where the data structure is relatively straightforward. However, more complex tasks such as object detection present greater challenges due to their intricate data structures (Everingham et al., 2015; Lin et al., 2014; Geiger et al., 2012; Sun et al., 2020). Only recently has a study (Lee et al., 2024) explored dataset pruning for object detection, but it offered limited investigation and achieved suboptimal performance. On the other hand, traditional dataset pruning methods, which assume full annotations, make a strong assumption that is not well suited for object detection, where annotation costs are high. Meanwhile, active learning techniques (Houlsby et al., 2011; Kirsch et al., 2019; Nguyen & Smeulders, 2004; Sener & Savarese, 2017; Agarwal et al., 2020; Xie et al., 2023; Li et al., 2023) prioritize selecting critical samples for labeling, which helps reduce the annotation burden. However, these methods often struggle to scale when applied to a large proportion of the full dataset, resulting in models that may not perform as well as those trained on the complete dataset (Wu et al., 2022; Yang et al., 2022). This limitation is especially concerning in precision-sensitive applications like autonomous driving, where safety and accuracy are paramount.

Therefore, in the context of object detection, we propose a novel paradigm that combines the advantages of data pruning with active learning to reduce annotation costs, as well as storage and training burdens. We term this paradigm **Annotation-Aware Data Pruning**. As shown in Fig. 1(b), this

Figure 1: A comparison between traditional data pruning and our Annotation-Aware Data Pruning. Traditional data pruning, depicted in (a), requires full annotations to determine which samples do not need to be trained. In the new paradigm, illustrated in (b), only the samples to be trained need labeling.

new paradigm requires labeling only the samples chosen for training, unlike traditional data pruning (Fig. 1(a)) that relies on full dataset annotations to identify removable samples.

To identify valuable samples without full annotations, we introduce a two-phase framework that integrates uncertainty estimation with redundancy removal to enable efficient data pruning and selection. In the first phase, referred to as **Initial Model Construction**, where the dataset lacks annotations, we aim to train a model capable of basic object detection. To achieve this, we select and annotate a small subset of the data to build an initial model. We then use uncertainty-based methods to identify and prioritize the most informative and challenging samples for further annotation. By calculating the uncertainty of each predicted object using entropy, and applying class-specific weights to account for class variations, we are able to rank and select the most uncertain samples for annotation. However, uncertainty alone can lead to redundant data selection, as similar objects may appear in different scenes. To address this, we apply a global object-level similarity metric to detect and remove redundant samples. By computing the cosine similarity between object features across scenes and using a threshold to assess redundancy, we ensure that the selected data is both informative and diverse. In the second phase, referred to as **Advanced Data Refinement**, we refine the model further by focusing on hard sample mining, dynamically adjusting class weights, and continuing to remove redundant samples. This two-phase process enables efficient data pruning, allowing the model to focus on challenging and valuable objects while minimizing redundancy, thereby improving both the dataset and model performance.

We evaluate our method on four datasets, including the 2D detection datasets PASCAL VOC and COCO, as well as the 3D detection datasets KITTI and Waymo. Our approach consistently achieves over a 20% pruning rate, greatly reducing annotation effort without any loss in performance.

To summarise, our contributions are as follows:

- We propose a novel **A**nnotation-**A**ware **D**ata **P**runing paradigm for object detection, dubbed as $A^2$-DP, which enables dataset pruning without requiring full annotations.

- We design a two-phase framework consisting of **Initial Model Construction** and **Advanced Data Refinement** to efficiently identify and annotate critical data.

- Extended experiments on 2D and 3D detection tasks validate the efficacy of our approach, consistently achieving a pruning rate of at least 20% across all tested datasets.

## 2 RELATED WORK

### 2.1 DATASET PRUNING

Dataset pruning refers to removing some samples from the whole dataset for model training. Recent studies have highlighted the potential for cost savings through data pruning. In works (Sorscher et al., 2022; Ayed & Hayou, 2023), researchers showcase that carefully designed pruning techniques

can yield results comparable with full datasets. Additionally, a study (Qin et al., 2023) employs loss to dynamically prune datasets, while another work (He et al., 2024) explores data pruning in large-scale settings. However, prior studies have predominantly focused on conducting research on dataset pruning within classification tasks, overlooking more intricate tasks such as object detection. Only recently, a study (Lee et al., 2024) has extended the dataset pruning task to object detection. This work generates image-wise and class-wise representative feature vectors to select samples that encapsulate both representativeness and diversity. However, the depth of its research is limited, with experiments conducted on a small portion of the data.

## 2.2 ACTIVE LEARNING

Active learning techniques have garnered considerable interest across diverse domains due to their capacity to mitigate the labeling workload. These methods can generally be classified into two main categories: uncertainty-based (Houlsby et al., 2011; Gal et al., 2017) and diversity-based strategies (Nguyen & Smeulders, 2004; Sener & Savarese, 2017; Agarwal et al., 2020; Xie et al., 2023; Li et al., 2023). Uncertainty-based strategies harness notions of uncertainty to pinpoint samples that offer the most informational value for annotation, thereby enhancing the learning process. On the other hand, diversity-based methodologies prioritize the selection of samples that encapsulate a wide range of characteristics and patterns present in the dataset, aiming to create a more comprehensive and representative training set. Moreover, recent studies (Huang et al., 2010; Ash et al., 2019) have delved into the fusion of uncertainty-based and diversity-based strategies, aiming to capitalize on the strengths of each approach. In recent years, there has been a surge of interest in the application of active learning techniques to object detection tasks (Yuan et al., 2021; Choi et al., 2021; Wu et al., 2022; Yang et al., 2022; Luo et al., 2023). Unlike image classification, active learning in object detection presents distinct challenges due to the intricacies involved in localizing and identifying objects within images. One method, ENMS (Wu et al., 2022) introduces entropy-based non-maximum suppression to assess uncertainty and explores various prototype strategies to ensure dataset diversity. Furthermore, PPAL (Yang et al., 2022) introduces a plug-and-play active learning methodology by considering both uncertainty and diversity. Nevertheless, existing methods predominantly concentrate on utilizing a limited dataset subset to represent the entire data, potentially resulting in models that may not be entirely comparable. Therefore, we believe there is a research gap concerning the utilization of a substantial percentage of labeled data.

## 3 METHODS

In this section, we present our comprehensive pipeline, which consists of two phases. In the first phase, our objective is to train a robust model capable of choosing crucial, challenging, and class-balanced samples from the entire dataset. Subsequently, in the second phase, we utilize the model developed in the first stage to uncover additional significant data beneficial for model training through rigorous data mining for hard samples and the removal of redundant data. The whole process can be seen in Pseudocode 1.

### 3.1 PROBLEM STATEMENT

Consider a dataset $D = \{d_{1...|D|} | d_i = (x_i, y_i)\}$, where $x_i$ denotes the input, such as images or LiDAR data, and $y_i$ represents the corresponding annotations. In object detection tasks, $y$ can represent class, location, orientation, and other relevant attributes. The objective of data pruning is to get a new dataset $D^- \subseteq D$ in such a way that

$$H = E_{z \sim P(D)}[L(z, \theta_D)] - E_{z \sim P(D)}[L(z, \theta_{D^-})] \tag{1}$$

approaches 0 (Yang et al., 2023), where $P(D)$ denotes the data distribution, z is a sample drawn from $P(D)$, $\theta_D$ denotes the model parameters trained on dataset $D$ and $L$ is a function that evaluates the performance of the model with parameters $\theta_D$ on $z$.

### 3.2 PHASE 1: INITIAL MODEL CONSTRUCTION

Given the absence of annotations for the datasets, we need a model to discern information such as object classes, locations, and features. Hence, we adopt active learning principles to develop an

---

**Algorithm 1** Pseudocode for the complete data selection and refinement process

---

**Input**: Unlabeled dataset $D = \{(x_i)\}_{i=1}^{|D|}$, initial model $M$, budget $B$, threshold $\tau$ for redundancy, class weights $W_c$.
**Output**: Pruned labeled dataset $D^-$
 1: **Phase 1: Initial Model Construction**
 2: Randomly select a subset $D_0 \subset D$ for annotation.
 3: Train initial model $M_0$ on labeled data $D_0$.
 4: **while** budget $B$ not exhausted **do**
 5:     Compute uncertainty $U(O)$ for each object $O$.
 6:     Compute class weights $W_c$ based on the labeled dataset.
 7:     Calculate the total uncertainty score for each data point $x$.
 8:     Sort the data points in descending order of uncertainty.
 9:     **for** each data point $x$ in sorted $D$ **do**
10:         Apply similarity-based redundancy removal.
11:         **if** data point $x$ is not redundant **then**
12:             Annotate $x$ and add to labeled dataset $D^-$.
13:         **end if**
14:     **end for**
15: **end while**
16: Update $M_0$ by retraining on the new $D^-$.
17: **Phase 2: Advanced Data Refinement**
18: **for** each input scene $x$ **do**
19:     Compute uncertainty $U(O)$ and Class weight $W_{c(O)}$ for each object $O$.
20:     Compute the aggregation weight for each object.
21:     Aggregate the total uncertainty for the scene.
22: **end for**
23: Apply similarity-based redundancy removal as in Phase 1.
24: Apply greedy selection as in Phase 1.
25: **return** Pruned labeled dataset $D^-$

---

effective model by selectively annotating a small subset of data. Initially, we randomly select a small subset of data for annotation and use this annotated data to train an initial model, enabling it to acquire a foundational ability for object detection. Subsequently, we utilize the initial model to identify essential data to enhance its detection capabilities. To be more specific, we employ uncertainty methods and similarity metrics for identification.

### 3.2.1 UNCERTAINTY-BASED SAMPLE SELECTION

To measure uncertainty, we utilize entropy as a key metric. Specifically, the uncertainty of a predicted object can be calculated as:

$$U(O) = \sum_{i=1}^{C} -p_i \log(p_i) \tag{2}$$

where $C$ is the number of classes, $O$ is the predicted object, and $p_i$ is the predicted probability of the object belonging to class $i$. However, during the first phase, the model's performance may be insufficient, resulting in numerous false positives that can distort the uncertainty assessment. To mitigate this, we apply a relatively high threshold to filter out incorrect predictions, ensuring that only valid objects are retained.

### 3.2.2 CLASS BALANCE ADJUSTMENT

In addition to uncertainty estimation, it is crucial to assign varying levels of attention to different classes. In this phase, we dynamically adjust the significance of each class based on the ratio of unlabeled data to training data for that class. We observed a strong correlation between this ratio

and performance improvements, as shown in Sec. 4.3.4. For each object, we further calculate the estimated improvement based on the number of unlabeled objects.

The weight for each class $c$, denoted as $W_c$, is computed by the number of predicted objects in the unlabeled data $D_{\text{unlabeled}}(c)$ and the number of objects in the training data $D_{\text{train}}(c)$:

$$W_c = \left( \frac{|D_{\text{unlabeled}}(c)|}{|D_{\text{train}}(c)|} \cdot \frac{1}{|D_{\text{unlabeled}}(c)|} \right)^{\alpha_1} = \left( \frac{1}{|D_{\text{train}}(c)|} \right)^{\alpha_1} \tag{3}$$

where $\alpha_1$ is a hyperparameter that prevents the weights from becoming excessively large or small. This weight ensures that classes with a lower number of training data receive more attention, allowing the model to prioritize classes that stand to benefit the most from additional labeling.

Finally, we aggregate the uncertainties of the predicted bounding boxes into a single uncertainty measure by averaging:

$$H(x) = \sum_{i=1}^{|O|} W_{c(O_i)} \times U(O_i)/|O| \tag{4}$$

where $x$ represents the input data (e.g., images or LiDAR data), $O$ denotes the set of predicted boxes for $x$, $|O|$ is the number of predicted boxes, and $c(O_i)$ is the class of the $i$-th predicted object.

After computing the uncertainty for each data point, we rank them in descending order and employ a greedy selection strategy to choose the most uncertain samples. To handle redundancy among the selected uncertain samples, we apply a similarity-based metric, which is described in the following section.

### 3.2.3 GLOBAL OBJECT-LEVEL SIMILARITY FOR REDUNDANCY ELIMINATION

In object detection tasks, comparing entire images independently often fails to efficiently capture redundant object instances that appear across different scenes. To address this, we propose a global object similarity module that operates at the object level, identifying and eliminating similar bounding boxes across multiple scenes. Once these object-level similarities are determined, they are aggregated to assess the overall redundancy of the image. As shown in Fig. 2(b), traditional image-wise similarity methods can only identify similar scenes and thus are limited in their ability to remove redundancy. In contrast, global object-level similarity can detect similar objects across different scenes, offering a more robust solution for redundancy elimination.

Let $F = \{f_1, f_2, ..., f_{|F|}\}$ represent the set of extracted object features for a given image, where each $f_i \in \mathbb{R}^d$ is a $d$-dimensional feature vector corresponding to a detected object. For a given class, we reduce the number of features using clustering algorithms such as KMeans to lower computational complexity when the feature set is large. Let $K$ be the number of clusters, and $C = \{c_1, c_2, ..., c_K\}$ denote the centroids of these clusters.

For each object feature $f_{\text{obj}}$ in the image, we compute the cosine similarity with every cluster centroid $c_k$:

$$\text{cosine\_similarity}(f_{\text{obj}}, c_k) = \frac{f_{\text{obj}} \cdot c_k}{\|f_{\text{obj}}\| \|c_k\|}, \quad \forall k \in \{1, ..., K\} \tag{5}$$

where $\cdot$ denotes the dot product, and $\|\cdot\|$ represents the Euclidean norm.

The similarity score for each object $f_{\text{obj}}$ is the maximum cosine similarity across all clusters:

$$S(f_{\text{obj}}) = \max_{k \in \{1, ..., K\}} \left( \text{cosine\_similarity}(f_{\text{obj}}, c_k) \right) \tag{6}$$

To evaluate whether an entire image is redundant, we aggregate the similarity scores of all the objects in the image. One approach is to compute the average similarity of all detected objects:

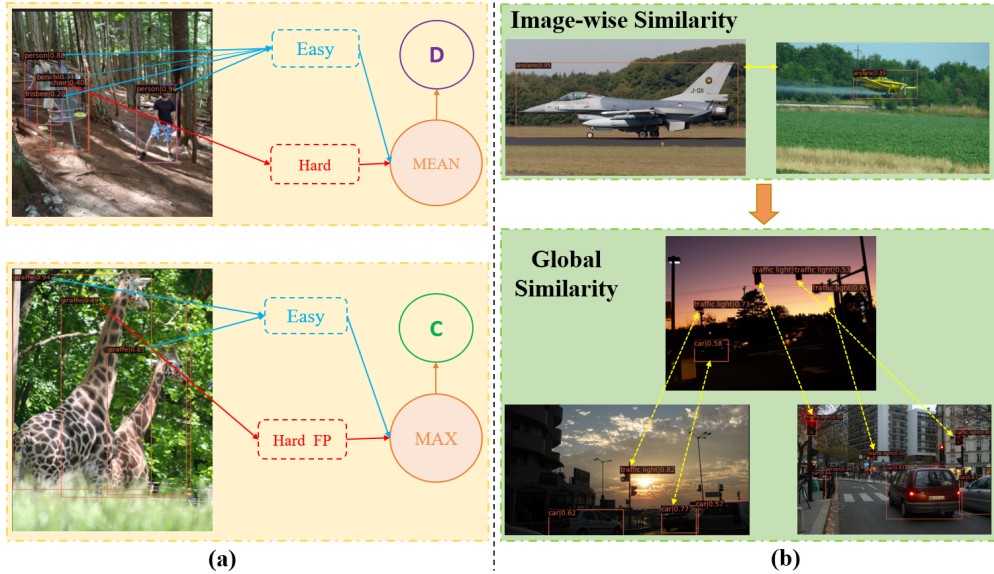

(a)                        (b)

Figure 2: Visualization of different uncertainty and similarity methods on the COCO dataset. Each predicted bounding box is displayed in red, along with its corresponding confidence score. The left part (a) illustrates the limitations of mean and max aggregation methods: max aggregation tends to choose(C) easy images by mistakenly highlighting false positives, while mean aggregation often disregards(D) hard samples due to the influence of numerous easy samples. The right part (b) compares different similarity methods. While image-wise similarity focuses on comparing entire images, which limits its ability to remove redundant objects, our global object-level similarity effectively identifies and removes redundant objects across different scenes.

$$S_{\text{image}} = \frac{1}{|F|} \sum_{i=1}^{|F|} S(f_i) \tag{7}$$

where $|F|$ is the number of objects detected in the image.

To decide whether to discard the entire image as redundant, we introduce a threshold $\tau$. If $S_{\text{image}} \geq \tau$, the image is considered redundant and is removed. This method ensures that an image is evaluated based on the aggregated similarity of its detected objects, allowing for the removal of images with redundant content.

The time complexity of the proposed global object-level similarity method involves several key steps. Feature extraction for each image with $|F|$ detected objects has a complexity of $O(|F|)$. Clustering the feature vectors using KMeans has a complexity of $O(T \cdot K \cdot d \cdot |F|)$, where $T$ is the number of iterations, $K$ is the number of clusters, $d$ is the feature dimension, and $|F|$ is the number of detected objects. The cosine similarity calculation between each object and the cluster centroids has a complexity of $O(|F| \cdot K \cdot d)$, and aggregating similarity scores and making redundancy decisions has a complexity of $O(|F|)$. Therefore, the total time complexity per image is $O(T \cdot K \cdot d \cdot |F| + |F| \cdot K \cdot d)$. For a dataset with $N$ images, this scales to $O(N \cdot |F| \cdot K \cdot d \cdot (T + 1))$, which is acceptable for large datasets, as the values of $T$, $K$, and $d$ are typically small constants.

### 3.3 PHASE 2: ADVANCED DATA REFINEMENT

After building a comprehensive model in the first phase, the second phase focuses on identifying additional critical data that may have been overlooked. While the first phase ensures sufficient representation of all objects, this phase emphasizes uncovering hidden but important objects through hard sample mining, class balance adjustment, and further redundant sample removal, similar to the methods employed in Phase 1.

### 3.3.1 HARD SAMPLE MINING

Identifying hard-to-detect yet important objects remains a significant challenge in object detection, particularly when these objects are overshadowed by simpler, easier-to-detect ones. As shown in the left-top of Fig. 2, hard samples are often surrounded by easy samples, resulting in a low overall image uncertainty, which can lead to their exclusion. To tackle this, we employ an advanced aggregation method that captures difficult samples more effectively, moving beyond traditional uncertainty-based approaches. Relying solely on the maximum uncertainty score can lead to biased results, as outliers or false positives may display high uncertainty and distort the selection process. For instance, hard false positives can significantly alter the uncertainty of entire images, causing incorrect image selections, as demonstrated in the left-bottom of Fig. 2. To address this, we propose a softmax-based aggregation method that prioritizes samples more equitably. This approach emphasizes the relative uncertainty of each object within a scene, ensuring a more balanced and accurate sample selection.

For a given input data $x$ with predicted bounding boxes $O$, we compute the uncertainty $U(o)$ for each predicted object $o$. Then, the aggregation weight $AW_i$ for object $i$ is calculated as:

$$AW_i = \frac{\exp(U(o_i))}{\sum_{j=1}^{|O|} \exp(U(o_j))} \tag{8}$$

Next, the total uncertainty score for the input scene $x$ is aggregated as:

$$H(x) = \sum_{i=1}^{|O|} W_{c(O_i)} \cdot U(O_i) \cdot AW_i \tag{9}$$

where $W_{c(O_i)}$ is the class-specific weight for object $O_i$, and $|O|$ is the total number of predicted objects in the scene.

## 4 EXPERIMENTS

### 4.1 EXPERIMENTAL SETTINGS

#### 4.1.1 DATASETS.

We evaluate the proposed method using three datasets: Pascal VOC (Everingham et al., 2015; 2010), COCO (Lin et al., 2014), KITTI (Geiger et al., 2012; 2013) and Waymo (Sun et al., 2020). Pascal VOC comprises 20 object categories. We utilize train2007+2012 for training and test2007 for testing. COCO is composed of 80 object categories, with 118,287 images and 860,001 annotated boxes. KITTI includes 3 object categories (Car, Pedestrian, Cyclist) with a training split consisting of 3,712 samples and a validation split containing 3,769 samples. For evaluation, we compute the mean average precision (mAP) at 40 recall positions for Car, Pedestrian (Ped), and Cyclist (Cyc), using 3D IoU thresholds of 0.7, 0.5, and 0.5, respectively, under moderate difficulty levels. Waymo consists of 798 training sequences and 202 validation sequences across three primary categories (Vehicle, Pedestrian, Cyclist). This extensive dataset contains over 10 million annotated objects. For simplicity, we consider 5% of the data as the full dataset. For evaluation, we compute the mean average precision (mAP) for Vehicle(Veh), Pedestrian (Ped), and Cyclist (Cyc) under Level_1 difficulty.

#### 4.1.2 IMPLEMENTATION DETAILS.

Following the previous active learning setting (Yang et al., 2022), we initially randomly sampled 5% of the data from the entire dataset for annotation and training the initial model. In Phase 1, we select an additional 40% of the data for model training within 4 rounds. In Phase 2, we conduct parallel experiments, selecting different portions of data (with a 10% increment between each proportion) to determine when the selected dataset closely approximates the full dataset. It is worth noting that the proportion of data is calculated based on the number of annotated boxes rather than images, as the detection performance is highly related with the number of objects (Lyu et al., 2023). Additionally,

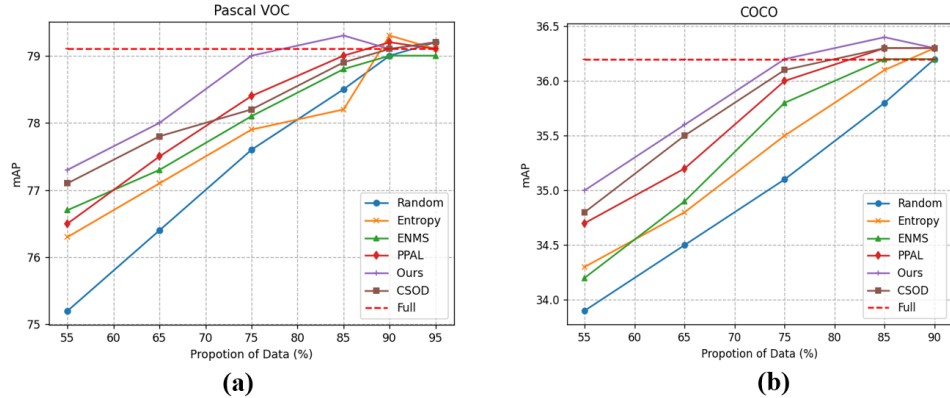

(a)           (b)

Figure 3: Comparison of results among different methods on the Pascal VOC dataset and COCO dataset across varying data proportions.

to ensure a fair comparison, we maintain a fixed seed and training iterations. Moreover, for the 2D detection task (Pascal VOC, COCO), we employ RetinaNet (Lin et al., 2017) as the detector, using the codebase from (Chen et al., 2019), whereas for the 3D object detection tasks (KITTI, Waymo), we utilize PV-RCNN (Shi et al., 2020) as the detector, with the codebase from (Team, 2020). The hyperparameter $\alpha_1$ in Class Balance Adjustment is set to 0.3, and the threshold $\tau$ in the Similarity module is set to 0.95.

### 4.1.3 COMPARED METHODS

Due to the absence of dedicated methods for dataset pruning in object detection, we primarily extend active learning techniques to handle larger data proportions, encompassing random sampling and traditional entropy methods. In the realm of 2D detection, we conduct comparative analyses with state-of-the-art active learning methods such as ENMS (Wu et al., 2022) and PPAL (Yang et al., 2022), alongside data pruning methods like CSOD (Lee et al., 2024), which leverage the complete annotations of the datasets. For the 3D task, we evaluate CRB (Luo et al., 2023), specifically designed for 3D object detection tasks. It is important to note that certain modules within these methodologies may entail significant time consumption. Therefore, we may adapt or omit them to ensure suitability for large-scale data settings.

### 4.2 EXPERIMENTAL RESULTS

### 4.2.1 ON PASCAL VOC

The results for each round on the Pascal VOC dataset are shown in Fig. 3(a). Our method achieves a pruning rate of 25%, utilizing only 75% of the data. In comparison, other methods manage only a 15% pruning rate, demonstrating the superior pruning efficiency of our approach. This highlights the limitation of many active learning methods, which perform well in low-data settings but struggle as data proportions increase, underscoring the importance of our research. Furthermore, our method outperforms CSOD, which relies on complete annotation information.

### 4.2.2 ON COCO

The results for each round on the COCO dataset are shown in Fig. 3(b). Similar to Pascal VOC, our method achieves a 25% pruning rate, utilizing only 75% of the data, whereas the best of other methods achieve a 20% pruning rate. This further emphasizes the increased pruning efficiency of our approach. Notably, our pruned results even surpass the performance of the full dataset, likely due to the presence of noisy labels in COCO. By removing these noisy labels, our method yields improved results.

Table 1: Comparison of results across methods on the KITTI dataset at 80% data, where our method first matches full dataset performance.

| Methods | Car | Ped | Cyc | Avg |
|---------|-----|-----|-----|-----|
| Random | 83.8 | 54.9 | 69.7 | 69.5 |
| Entropy | 84.1 | 56.0 | 71.2 | 70.4 |
| CRB | 83.7 | 56.9 | 72.2 | 70.9 |
| Ours | **84.3** | **57.5** | **72.6** | **71.4** |
| Full | 84.4 | 57.6 | 72.4 | 71.5 |

Table 2: Comparison of results across methods on the Waymo dataset at 70% data, where our method first matches full dataset performance.

| Methods | Veh | Ped | Cyc | Avg |
|---------|-----|-----|-----|-----|
| Random | 71.0 | 65.2 | 63.6 | 66.6 |
| Entropy | **71.1** | 65.8 | 63.2 | 66.7 |
| CRB | 70.9 | 66.8 | 62.8 | 66.8 |
| Ours | **71.1** | **66.9** | **64.1** | **67.4** |
| Full | 71.0 | 66.5 | 63.3 | 66.9 |

### 4.2.3 ON KITTI

The detailed results for the KITTI dataset, presented in Tab. 1, focus on the data proportion of 80%, which is the point where our method first matches the performance of the full dataset. Our method delivers comparable results across all categories using only 80% of the data, while other methods require approximately 90% of the data to achieve similar performance, and still fall short in certain categories.

### 4.2.4 ON WAYMO

The final results for the Waymo dataset are presented in Tab. 2, with a data proportion of 70%. Our method outperforms other approaches, even surpassing the results of the full dataset, particularly in challenging classes such as cyclists and pedestrians. This improvement is largely due to our uncertainty and class balance modules, which prioritize difficult samples and underrepresented classes, thereby improving the overall effectiveness of the training process.

### 4.3 ANALYSIS

Table 3: Ablation study of all components of our methods on the Pascal VOC dataset with a data proportion of 75%.

| Uncertainty | Class Balance | Similarty | mAP |
|-------------|---------------|-----------|-----|
| - | - | - | 77.3 |
| ✓ | - | - | 78.2 |
| - | ✓ | - | 78.3 |
| - | - | ✓ | 77.8 |
| ✓ | ✓ | - | 78.8 |
| ✓ | ✓ | ✓ | 79.0 |

Table 4: Ablation study of different uncertainty aggregation methods and similarity components of our methods on the Pascal VOC dataset with a data proportion of 75%.

| Uncertainty | Similarty | mAP |
|-------------|-----------|-----|
| Mean | Global | 78.6 |
| Sum | Global | 78.4 |
| Max | Global | 78.7 |
| Softmax | Pair-wise | 78.1 |
| Softmax | Divproto | 78.5 |
| Softmax | PPAL | 78.8 |
| Softmax | Global | 79.0 |

### 4.3.1 ABALATION ON UNCERTAIN, CLASS BALANCE AND SIMILARITY IN PHASE 2

As indicated in Tab. 3, we carry out experiments on each module of our methods. All modules, comprising uncertainty, class balance, and similarity, contribute significantly to the overall results. Particularly, the class balance module plays a crucial role in identifying rare classes, thereby reducing the disparity between the pruned dataset and the full dataset. The synergy among these modules further enhances the ultimate performance.

### 4.3.2 ABALATION ON AGGREGATION METHODS

In Tab. 4, we compare various uncertainty aggregation methods, such as mean, sum, max, and softmax. The results indicate that softmax aggregation is more effective in extracting hidden important objects compared to the other methods. This is because mean and sum aggregations can be influenced by other concurrently present simple objects, while max aggregation can be signifi-

cantly impacted by uncertain false positives. Only softmax aggregation can effectively extract the genuinely important objects while mitigating the negative effects of false positives

### 4.3.3 ABALATION ON SIMILARITY METHODS

In Tab. 4, we compare different similarity methods, including Pair-wise (Pair-wise comparison), Divproto (Wu et al., 2022), PPAL (Yang et al., 2022), and Global(our global similarity). The results indicate that our global similarity exhibits superior redundant removal capabilities. While other methods are confined to pair-wise comparisons, our approach starts with identifying objects that bear similarities across various scenes.

### 4.3.4 ANALYSIS ON CLASS BALANCE ADJUSTMENT

Class balance adjustment plays a crucial role, as we observe that the performance gap between the full data model and the pruned data model primarily stems from a few specific classes, while the performance of other classes remains comparable. To better demonstrate the impact of class balance adjustment, we evaluate different methods for calculating class weights on the COCO dataset, which has 80 classes—significantly more than other datasets. As shown in Tab. 5, we conducted experiments to analyze the correlation between AP differences of the full data model and the pruned data model by class and various class weight calculation methods. LDP method has the highest correlation coefficient (0.62) with the AP differences across classes. This suggests that our method of weighting classes based on the proportion of labeled data relative to the total data is more effective in capturing the actual class weights. In contrast, the average confidence score (Score) and average uncertainty (Ent) methods also exhibit low correlations (0.18 and 0.20, respectively). This implies that these metrics are less predictive of the AP differences and may not be as effective for class weighting.

Table 5: Correlation coefficients between AP differences and class weighting methods on the COCO dataset. LDP (Labeled Data Proportion) represents our method, which is the proportion of labeled data relative to the total data. Score indicates the average confidence score for each class on the unlabeled data, while Ent represents the average uncertainty (entropy) for each class on the unlabeled data.

| Class Weighting Method | LDP | Score | Ent |
|---|---|---|---|
| Correlation Coefficient | 0.62 | 0.18 | 0.20 |

### 4.3.5 COMPARISON BETWEEN FULL DATA MODEL AND INITIAL MODEL

Traditional data pruning requires fully labeled datasets to train the model and understand the overall data distribution. In contrast, since we lack complete annotations, we train an initial model on a labeled subset in Phase 1 to approximate the full dataset. Here, we compare the two approaches. First, the mAP of the full data model and the initial model are relatively close on the COCO test set (36.7 vs 33.6) and on the training set (44.1 vs 40.2), indicating that the initial model effectively mimics the performance of the full dataset. Additionally, the initial model is capable of accurately estimating object counts, making it well-suited for our class balance module. Moreover, uncertainty samples selected by the initial model tend to correspond to high-loss samples, which are valuable for training. On the other hand, uncertainty samples from the full data model may include noisy labels (Swayamdipta et al., 2020), which can negatively impact training.

## 5 CONCLUSION

In this paper, we present Annotation-Aware Data Pruning for object detection, $A^2$-DP, which reduces both the annotation and training burdens. Our approach features a two-phase pruning process that integrates a hard sample mining module to extract vital but concealed objects and a class balance module to identify significant objects in rare or challenging classes. Additionally, our global similarity removal module distinguishes our method by enabling object-level similarity assessments, improving the elimination of redundant information. Extensive experiments on 2D and 3D detection tasks demonstrate the effectiveness of $A^2$-DP, consistently achieving a pruning rate of at least 20% across various datasets, highlighting the practical value and efficiency of our approach.

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

# A Appendix

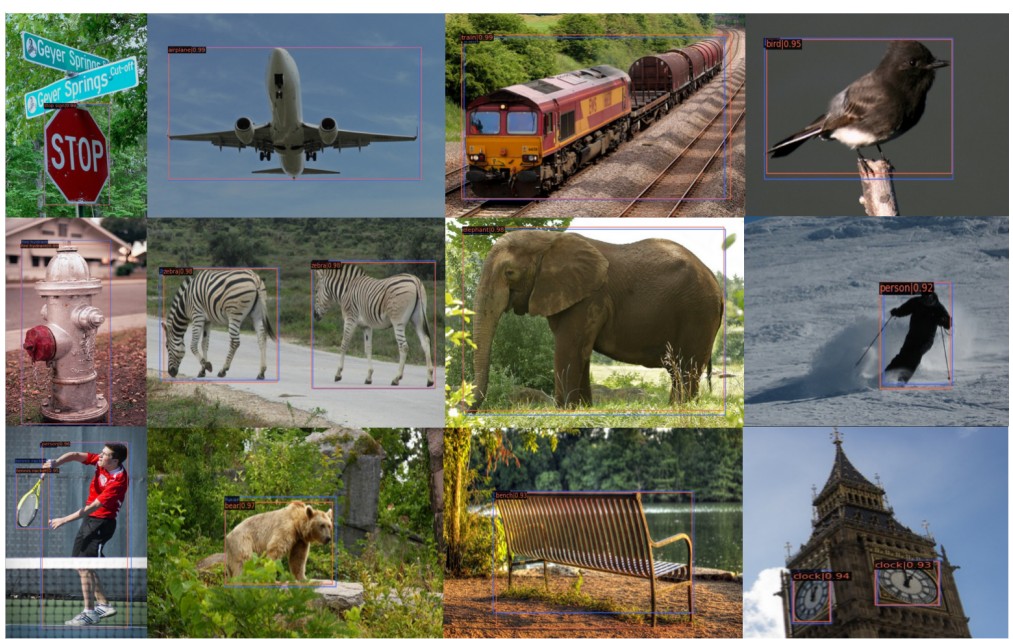

Figure 4: Visualization of **easy** samples in the COCO dataset. Red boxes represent predicted bounding boxes with their corresponding confidence scores, while blue boxes indicate GT boxes.

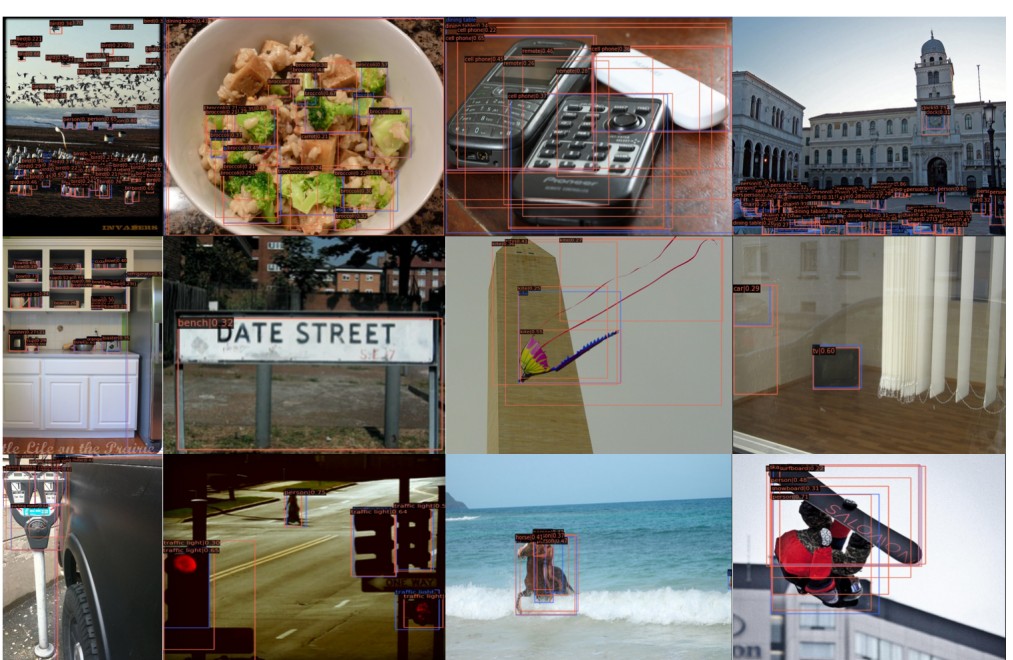

Figure 5: Visualization of **hard** samples in COCO. Red boxes represent predicted bounding boxes with their corresponding confidence scores, while blue boxes indicate GT boxes.

