# OpenReview forum: "$A^2$-DP: Annotation-aware Data Pruning for Object Detection"
_ICLR.cc/2025/Conference — Submitted to ICLR 2025_

### Official Review · Reviewer_HYeX · 2024-10-19

**Soundness:** 3
**Presentation:** 2
**Contribution:** 3
**Rating:** 3
**Confidence:** 5

**Summary:**

The article presents a novel Annotation-Aware Data Pruning (A²-DP) method aimed at reducing the annotation and storage costs in object detection tasks. The method consists of two phases: Initial Model Construction: This phase involves training a basic object detection model without complete annotations by utilizing techniques such as hard sample mining, class balance adjustment, and global similarity removal. Advanced Data Refinement: This phase focuses on mining hard-to-detect key samples from the previously constructed model and further eliminating redundant data. Finally, the authors provide extensive experiments on both 2D and 3D detection tasks to show the effectiveness of the propsoed method.

**Strengths:**

- The topic is meaningful. Labelled data, especially the 3D labelled,  is kind of valuable resouces for the current deep learning community. This work provide a data pruning method for both 2D/3D object detection.

- The proposed method is reaonable.

**Weaknesses:**

- Limited experiments and inappropriate/unclear experimental setting. It is not clear that what detectors/backbones/experimental settings are used in the experiments. Current numbers are significantly lower than the performance of SOTA methods, and can not support the claims and conclusions. To demonstrate the effectiveness of the proposed method, the author should provide the experimental results of **different object detectors and different backbones with and without the proposed method**. The current experiments are ambiguous and cannot draw effective conclusions。

- Inappropriate ablation study. VOC is out-of-dated dataset for object detection, and the ablation should be conducted on COCO dataset.

- The presentation should be further improved.

Overall, this work proposed a data pruning method for both 2D/3D dataset. Although the task is meaningful and the solution is reasonable, the experimental result can not support the claims and conclusions. Based on this, I tend to reject this submission.

**Questions:**

Please see the weaknesses.

---

> ### Author Response · Authors · 2024-11-21
> **Response to Reviewer HYeX**
>
> Dear Reviewer HYeX,
>
> We sincerely appreciate your thoughtful comments and valuable suggestions on our work.
>
> We are currently **conducting additional experiments** to address your feedback thoroughly. As such, we may need a bit more time to provide a detailed reply. Thank you for your understanding and patience.

---

> ### Author Response · Authors · 2024-11-25
> **Response to Reviewer HYeX(Part 2)**
>
> Dear Reviewer HYeX,
>
> We sincerely appreciate your review with thoughtful comments and valuable feedbacks. We have carefully considered each of your questions and provide detailed responses below. Please let us know if you have any further questions or concerns.
>
> **Q1: More  experimental results.**
>
> A1:  Sorry for missing details about experiments. In the paper, we use Retinanet[2] detector with ResNet50 backbone for 2D detection task and PV-RCNN detector with voxel-based backbone for 3D detection.
>
> Thank you for pointing out the need for more details and comprehensive experimental settings. We apologize for the lack of clarity in the initial version of the manuscript. Below are the updated experimental settings and additional results to validate our method across a wider range of detectors and backbones.
>
> In our experiments:
>
> - **For 2D object detection**, we used **RetinaNet** [2] with ResNet50 backbone, as mentioned in the original submission. In addition to this, we now include results for **DINO (R50)** [1], **RetinaNet (R101)**, and **ATSS (R101)** [3] with 75% of the data to demonstrate that our method works effectively across multiple detector-backbone combinations. These detectors are widely recognized baselines in active learning and data pruning research.
>
>   Table I: COCO Average mAP
>
>   | Settings        | W/o ours | W/ ours | Full |
>   | --------------- | -------- | ------- | ---- |
>   | DINO(R50)       | 48.2     | 49.0    | 48.9 |
>   | Retinanet(R101) | 37.9     | 38.5    | 38.7 |
>   | ATSS(R101)      | 40.2     | 40.7    | 40.8 |
>
> - **For 3D object detection**, we originally used **PV-RCNN** with a voxel-based backbone. To further validate our method, we now include results for **SECOND** [4] and **PointPillar** [5], which are strong baselines in 3D object detection tasks.
>
>   Table II: KITTI Average 3D Moderate mAP
>
>   | Settings    | W/o ours | W/ ours | Full |
>   | ----------- | -------- | ------- | ---- |
>   | SECOND      | 65.1     | 66.0    | 66.1 |
>   | PointPillar | 63.6     | 64.9    | 64.8 |
>
> These additional results demonstrate that our method consistently improves performance across different detectors and backbones for both 2D and 3D object detection tasks.
>
> We will include these results in the final version of the manuscript to provide more comprehensive validation and transparency.
>
> [1] DINO: DETR with Improved DeNoising Anchor Boxes for End-to-End Object Detection. ICLR 2023.
>
> [2] Focal Loss for Dense Object Detection. CVPR 2017.
>
> [3] Bridging the Gap Between Anchor-based and Anchor-free Detection via Adaptive Training Sample Selection. CVPR 2020.
>
> [4] SECOND: Sparsely Embedded Convolutional Detection. Sensors, 2012.
>
> [5] PointPillars: Fast Encoders for Object Detection from Point Clouds. CVPR 2018
>
>
>
> **Q2: Ablation study on COCO.**
>
> A2: Thank you for your suggestion regarding an ablation study. We conducted additional experiments on the COCO dataset using RetinaNet (R50), complementing the results from VOC in our paper. These findings provide further validation of the contributions of each component in our method. The updated results will be included in the final version of the paper.
>
> Table III: Ablation Study Results on COCO Dataset (mAP).
>
> | Uncertainty | Class Balance | Similarty | mAP  |
> | ----------- | ------------- | --------- | ---- |
> | -           | -             | -         | 35.1 |
> | √           | -             | -         | 35.5 |
> | -           | √             | -         | 35.7 |
> | -           | -             | √         | 35.4 |
> | √           | √             | -         | 36.1 |
> | √           | √             | √         | 36.3 |
>
> These results highlight the individual and combined contributions of uncertainty, class balance, and similarity. Notably:
>
> - **Uncertainty** and **class balance** independently improve performance, showing mAP gains of +0.4 and +0.6, respectively.
> - **Similarity** contributes to an additional boost, particularly when combined with the other two components.
> - The full method achieves the highest mAP of 36.3, demonstrating the synergy between these components.
>
> We appreciate your feedback and look forward to incorporating these findings into the final manuscript. Let us know if further clarification is needed!
>
> **Q3: The presentation.**
>
> A3: We appreciate your suggestions for improving the presentation of our paper. We will revise our paper carefully.

---

### Official Review · Reviewer_Sqcz · 2024-11-03

**Soundness:** 4
**Presentation:** 3
**Contribution:** 3
**Rating:** 6
**Confidence:** 4

**Summary:**

This paper has two main phases: initial model training and advanced data refinement. In the first phase, a portion of the data is annotated and used to train an initial model, which then selects the most informative samples based on uncertainty and class balance. Redundant samples are further removed using object-level similarity analysis. The second phase further refines the data using hard sample mining, class balance optimization, and redundancy reduction.
So this paper:
1.Introduced A2-DP, a method that effectively reduces data and annotation volume for object detection without compromising performance.
2.Used entropy-based uncertainty and object-level similarity to choose informative samples and eliminate redundant data.
3.Showed significant pruning rates (at least 20%) across various datasets, demonstrating A2-DP's practical benefits in reducing data requirements for training high-performance models.

**Strengths:**

1.A2-DP adjusts its pruning strategy based on the data, using both uncertainty and class balance, ensuring that the model focuses on the most valuable data, leading to improved generalization.
2.The authors validated A2-DP on multiple datasets, including both 2D and 3D object detection, achieving a consistent pruning rate of at least 20%. This scalability shows the method's generalizability across different applications and data types.

**Weaknesses:**

1.The method requires additional training for the initial model, which could increase training costs. The authors should clarify the exact training cost for the initial model. Additionally, an ablation study should be added to compare their method with other advanced methods under the same training cost conditions.
2.In line 502, "conducted" should be replaced with "conduct."
3.The explanation of how the budget parameter (B) works, along with the threshold parameter, is unclear in the algorithm and should be elaborated.

**Questions:**

refer to the weakness for the rebuttal

---

> ### Author Response · Authors · 2024-11-21
> **Response to Reviewer Sqcz**
>
> Dear Reviewer Sqcz,
>
> We sincerely appreciate your review with thoughtful comments and positive feedbacks. We have carefully considered each of your questions and provide detailed responses below. Please let us know if you have any further questions or concerns.
>
> **Q1: The clarification about training cost for the initial model.**
>
> A1: Thank you for your suggestion. We focused less on training costs because the primary expense in object detection typically comes from annotation, and the cost of training the initial model is significantly lower than the cost of wasted annotations.
>
> Additionally, there may be a misunderstanding. The training of the initial model is necessary for this task. Traditional data pruning methods often train an initial model using the full dataset with complete ground truth labels, whereas we only use a portion of the data, significantly reducing annotation costs. In this case, our training cost is lower than that of traditional data pruning methods. Furthermore, when comparing our method with other active learning methods, the training costs are comparable.
>
> **Q2: Typos.**
>
> A2: We appreciate your suggestions for improving the presentation of our paper. We will revise our paper carefully.
>
> **Q3: The explanation of the parameters.**
>
> A3:  The budget parameter (B) is set with a 10% increment between each proportion to balance time and computational resource constraints. Our experiments demonstrate that at least 20% of the labeled data is redundant in common object detection datasets。 The threshold parameter is determined empirically based on preliminary experiments. We will clarify these details in the revised manuscript to improve the algorithm's explanation.

---

### Official Review · Reviewer_VpmP · 2024-11-03

**Soundness:** 2
**Presentation:** 2
**Contribution:** 2
**Rating:** 5
**Confidence:** 3

**Summary:**

This paper introduces Annotation-Aware Data Pruning ($A^{2}\text{-}DP$), an approach for object detection aimed at reducing annotation costs and storage requirements while maintaining model performance. $A^{2}\text{-}DP$ leverages a hard sample mining module to capture hidden objects, a class balance module to prioritize rare or challenging classes and a global similarity removal module to eliminate redundant data through object-level similarity analysis. Experiments on both 2D and 3D detection tasks demonstrate at least a 20% pruning rate.

**Strengths:**

- Dataset pruning for object detection addresses an important but under-explored area in deep learning.

- The approach of pruning datasets without needing full annotations is impressive and offers practical advantages.

**Weaknesses:**

- The technical novelty of this paper seems limited, as several applied modules and concepts—such as uncertainty, class-wise, and object-level learning—have already been explored in [1, 2]. This paper appears to primarily extend [1] to a wider range of datasets, including 3D object detection. However, the reviewer may be overlooking some unique aspects and welcomes clarification on this point.

[1] Lee, Hojun, et al. "Coreset Selection for Object Detection." Proceedings of the IEEE/CVF Conference on Computer Vision and Pattern Recognition. 2024.
[2] He, Muyang, et al. "Large-scale dataset pruning with dynamic uncertainty." Proceedings of the IEEE/CVF Conference on Computer Vision and Pattern Recognition. 2024.

- In Section 3.3.1, the authors propose a softmax-based aggregation method to address hard false positives in their hard sample mining scheme. However, it’s unclear how softmax uncertainty scores would address this issue effectively, as they might increase the probability distribution weighting toward the negative class. A visual illustration demonstrating this approach’s effectiveness could clarify its impact.

**Questions:**

Since one of the main claims of this work is to reduce the need for a fully annotated dataset for pruning, is it appropriate to compare this approach directly with semi-supervised object detection methods [3, 4]? Moreover, how this method compares to semi-supervised approaches in terms of annotation requirements and computational efficiency. What are the potential advantages or disadvantages of the proposed method compared to semi-supervised techniques for object detection?

[3] Xu, Mengde, et al. "End-to-end semi-supervised object detection with soft teacher." Proceedings of the IEEE/CVF international conference on computer vision. 2021.

[4] Zhang, Jiacheng, et al. "Semi-detr: Semi-supervised object detection with detection transformers." Proceedings of the IEEE/CVF conference on computer vision and pattern recognition. 2023.

---

> ### Author Response · Authors · 2024-11-21
> **Response to Reviewer VpmP (Part 1)**
>
> Dear Reviewer VpmP,
>
> We sincerely appreciate your review with thoughtful comments. We have carefully considered each of your questions and provide detailed responses below. Please let us know if you have any further questions or concerns.
>
> **Q1: The novelty of the paper.**
>
> A1: While our method utilizes uncertainty and class-balanced strategies, these are implemented in unique ways, making them less common in the context of data selection for object detection:
>
> - **Uncertainty Perspective**: Our contribution lies in the **aggregation of object-level uncertainty**, a novel approach that ensures adaptability and compatibility with future advancements in uncertainty estimation methods. This aggregation strategy is specifically tailored for object detection, which involves more complex data structures than traditional classification tasks.
> - **Diversity Perspective**: We propose a specific diversity-related innovation: **a global object-level similarity** metric to effectively remove redundant samples across scenes, enhancing the dataset's representativeness. This approach goes beyond standard pairwise similarity methods and addresses redundancy at the object level, a key challenge in object detection tasks.
> - **Class Balance Perspective**: Our method introduces **dynamic class-balanced weighting**, a simple yet effective strategy that prioritizes underrepresented or harder classes during sample selection. This adjustment significantly improves the performance of pruned datasets, as demonstrated in our experiments.
>
> These contributions collectively address key challenges in data selection for object detection, ensuring both practical utility and theoretical novelty.
>
> Comparison with CSOD [1] highlights several key differences: our method introduces unique techniques in uncertainty aggregation, similarity measurement, and class balance adjustment. Additionally, CSOD relies on GT information, making their process fundamentally different from ours. Moreover, as shown in Fig. 3, our method outperforms CSOD even without leveraging GT information, demonstrating the effectiveness of our approach. Regarding [2], it focuses on image-level tasks, which do not address the unique challenges of object detection, such as handling multiple objects and complex backgrounds within a single scene. This distinction underscores the specific contributions and relevance of our approach to object detection tasks.
>
> We will clarify these aspects in the revised manuscript.
>
> **Q2: The clarification** **about the softmax-based aggregation method.**
>
> A2: Thank you for your feedback. The softmax-based aggregation method is designed to address the issue of FPs effectively while emphasizing hard objects during training. To account for the varying effects of aggregation methods, we adapt them dynamically across different phases:
>
> **Uncertainty Dynamics Across Phases**:
>
> - **In Phase 1**, when the data volume is smaller and many objects are yet to be labeled, we use **average uncertainty aggregation** to ensure the selected objects are not overly influenced by uncertain FPs. This balances exploration when the dataset is diverse.
> - **In Phase 2**, as the remaining objects are fewer and consist primarily of simple objects and hidden hard samples, the focus shifts to emphasizing hidden hard objects, even when surrounded by simpler ones. For this, **softmax aggregation** is employed, as it prioritizes objects with higher uncertainty while mitigating the impact of uncertain FPs. This approach is more robust than max aggregation, which can be overly sensitive to outliers.
>
> **Visual Illustration**:
>
> - The effectiveness of this approach can be seen in Figure 2(a). The top example contains one hard sample surrounded by many simpler objects; in such cases, average aggregation is likely to overlook the hard sample, whereas max and softmax methods can successfully identify and prioritize it. The bottom example, on the other hand, demonstrates that max aggregation introduces excessive noise due to the influence of FPs, while softmax effectively mitigates this issue, providing a more balanced selection.
>
> We will further elaborate on this in the revised manuscript to clarify the method's impact.

---

> ### Author Response · Authors · 2024-11-21
> **Response to Reviewer VpmP (Part 2)**
>
> **Q3: The relationship between semi-supervised methods and our method.**
>
> A3: Our method complements semi-supervised approaches by identifying and labeling hard samples while leaving easy samples unlabeled, making the remaining data suitable for semi-supervised learning. This hybrid approach reduces annotation costs by focusing on the most informative data, while leveraging semi-supervised learning to handle the rest efficiently. Compared to purely semi-supervised methods, our approach offers better control over the labeled datasets' quality and ensures that the most critical samples are accurately labeled, potentially improving model performance in challenging scenarios.
>
> Additionally, semi-supervised learning often relies on pseudo-labels, which can introduce noise and hinder the model's ability to achieve performance close to that of using fully labeled data. By reducing reliance on pseudo-labels, our method mitigates these risks and ensures a more robust training process.

---

### Official Review · Reviewer_uQgi · 2024-11-04

**Soundness:** 3
**Presentation:** 2
**Contribution:** 2
**Rating:** 5
**Confidence:** 4

**Summary:**

This paper introduces A2-DP, an Annotation-Aware Data Pruning paradigm for object detection, which aims to reduce the annotation and storage burdens associated with large datasets. A2-DP employs a two-phase framework that includes hard sample mining to extract critical hidden objects, a class balance module to identify important objects in rare or challenging classes, and a global similarity removal module to eliminate redundant information through object-level similarity assessments. The method has been extensively tested on 2D and 3D detection tasks, consistently achieving a minimum pruning rate of 20% across various datasets without compromising performance, demonstrating its practical value and efficiency.

**Strengths:**

1. This paper focuses on data efficiency for object detection, which is very important for current data-scarce computer vision tasks because the annotations and storage for object detection are both costly and burdensome.
2. The proposed method seems to be effective under different pruning rate, and has been deomistreted on both 2D and 3D object detection benchmarks.
3. To evaluate the similarity in object level is reasonable because the object detection model typically focuses more on objects instead of scenes.

**Weaknesses:**

1. The definition of the problem in this work is not clear enough. Although the authors attempt to use the concepts of data pruning, the problem is an active learning problem. The only difference between them is that the portion of labeled samples in active learning is smaller than that in the proposed method. Therefore, there is no essence difference between them. Moreover, the problem statement in section 3.1 is also incomplete in describing the problem of this paper.
2. Lack of novelty. The proposed method leverages the uncertainty and class-balanced weights are very common in the data selection for object detection.
3. The presentation needs to be improved. It would be better to switch the order for Figure 2b and Figure 2a according to their reference in the main text. Some math definitions are also not well defined, e.g., the $O$, $o$, and $O_i$.
4. As the paper states, the proportion of data is calculated based on the number of annotated boxes rather than images. The authors may provide specific descriptions of the procedure. The images usually contain different numbers of objects, how to give an exact 20% data according to the number of objects?

**Questions:**

1. Any other state-of-the-art active learning methods be used for comparison? The compared methods seem to be slightly outdated for the year 2024.
2. It seems that the model trained on only 80% COCO data can outperform that trained on the full data. What is the reason behind this? Does it mean the redundancy data in COCO would have a negative impact on the model?

---

> ### Author Response · Authors · 2024-11-21
> **Response to Reviewer uQgi**
>
> Dear Reviewer uQgi,
>
> We sincerely appreciate your review with thoughtful comments and valuable suggestions. We have carefully considered each of your questions and provide detailed responses below. Please let us know if you have any further questions or concerns.
>
> **Q1: The difference our A2-DP and AL.**
>
> A1: Thank you for your insightful question. There are clear differences between our A2-DP and traditional active learning:
>
> 1. **Goal**: A2-DP focuses on maintaining performance without degradation, particularly in high-risk scenarios where consistency is critical. This makes it more suitable for real-world applications.
> 2. **Methodology**: Traditional active learning typically selects a small proportion of samples intended to represent the entire dataset. In contrast, A2-DP aims to exclude unimportant samples from large-scale datasets, which better aligns with scenarios involving large data proportions. Our experiments further show that traditional active learning methods often fail to achieve comparable results to models trained on the full dataset and sometimes perform similarly to random pruning.
>
> **Strengths of A2-DP**: A2-DP combines the strengths of both paradigms, effectively handling large data proportions while minimizing additional annotation costs.
>
> We appreciate your suggestion and will revise Section 3.1 to include this explanation, clarifying both the differences and strengths of our approach.
>
> **Q2: The novelty of our methods.**
>
> A2: While our method utilizes uncertainty and class-balanced strategies, these are implemented in unique ways, making them less common in the context of data selection for object detection:
>
> - **Uncertainty Perspective**: Our contribution lies in the **aggregation of object-level uncertainty**, a novel approach that ensures adaptability and compatibility with future advancements in uncertainty estimation methods. This aggregation strategy is specifically tailored for object detection, which involves more complex data structures than traditional classification tasks.
> - **Diversity Perspective**: We propose a specific diversity-related innovation: **a global object-level similarity** metric to effectively remove redundant samples across scenes, enhancing the dataset's representativeness. This approach goes beyond standard pairwise similarity methods and addresses redundancy at the object level, a key challenge in object detection tasks.
> - **Class Balance Perspective**: Our method introduces **dynamic class-balanced weighting**, a simple yet effective strategy that prioritizes underrepresented or harder classes during sample selection. This adjustment significantly improves the performance of pruned datasets, as demonstrated in our experiments.
>
> These contributions collectively address key challenges in data selection for object detection, ensuring both practical utility and theoretical novelty. We will clarify these aspects in the revised manuscript.
>
> **Q3: Revision about the presentation.**
>
> A3: We appreciate your suggestions for improving the presentation of our paper. We will carefully revise the figure order, clarify all mathematical definitions, and improve the problem statement.
>
> **Q4: The clarification of the pruning procedure.**
>
> A4: Sorry for missing this description to select data.  We use GT information to ensure the pruning rate is exactly 20%. Specifically, our pruning process is greedy, and we stop pruning once more than 20% of objects have been removed. Since individual scenes contain relatively few objects compared to the entire dataset, the potential bias is negligible. Importantly, GT information is only used during this step to ensure a fair comparison across methods.
>
> **Q5: Comparison with SOTA AL methods.**
>
> A5: We acknowledge the importance of comparing our method with SOTA AL approaches. However, recent advancements in active learning for object detection are limited.
>
> The most notable recent work is KECOR[1]. We have included a comparison with KECOR in our experiments to provide a comprehensive evaluation of our method's performance.
>
> | KITTI | Car  | Ped  | Cyc  | Avg  |
> | ----- | ---- | ---- | ---- | ---- |
> | KECOR | 84.1 | 57.2 | 72.0 | 71.1 |
> | Ours  | 84.3 | 57.5 | 72.6 | 71.4 |
>
> | Waymo | Veh  | Ped  | Cyc  | Avg  |
> | ----- | ---- | ---- | ---- | ---- |
> | KECOR | 71.1 | 66.8 | 63.5 | 67.1 |
> | Ours  | 71.1 | 66.9 | 64.1 | 67.4 |
>
> [1] KECOR: Kernel Coding Rate Maximization for Active 3D Object Detection. ICCV 2023
>
> **Q6: The explaination about the performance on COCO.**
>
> A6: The improved performance on 80% COCO data is likely due to a combination of factors:
>
> (1) The pruning process may reduce mislabeled samples present in the COCO dataset, leading to cleaner data; (2) it eliminates simple samples, enabling the model to focus on harder, more informative examples; (3) and it adjusts the class distribution, allowing the model to emphasize underrepresented or challenging classes. Together, these factors enhance the model's learning.

---

> > ### Comment · Reviewer_uQgi · 2024-11-24
> > **Response to the authors**
> >
> > Thanks for the response from the authors. Most of my concerns have been addressed. However, the novelty of this paper is still limited. As the authors state, the uncertainty is an aggregation of object-level uncertainty, but it is usually used to evaluate the image uncertainty by using the object-level uncertainty. The class balance with dynamic class-balanced weighting is also a common technique for addressing the class imbalance issue in object detection.
> >
> > Most importantly, the authors provide a comparison of the experiments between the proposed method and the SOTA active learning methods, we can only observe minimal performance improvements (0.3% AP), which significantly determines the effectiveness of the proposed method. It further reduces the difference between the active learning and the proposed method. Therefore, I tend to keep my score unchanged.

---

> > > ### Author Response · Authors · 2024-11-24
> > > **Clarification of Novelty and Improvements**
> > >
> > > Thank you for your thoughtful comments and for sharing your concerns.
> > >
> > > Regarding **novelty**, in object detection, the presence of multiple objects and complex backgrounds in a single scene makes image uncertainty fundamentally different from object-level uncertainty. **Aggregating uncertainty effectively is critical**, as our experiments demonstrate that different aggregation methods lead to significantly different outcomes. While dynamic class-balanced weighting is a common technique, **determining what metrics to use for weighting in object detection is non-trivial and requires substantial effort**. Research in this area, especially in the context of object detection, should not be overlooked.
> > >
> > > From a broader perspective, we are the first to **explore data pruning while explicitly considering annotation costs in object detection tasks**, a topic of paramount importance, particularly in high-risk scenarios such as autonomous driving. Moreover, our method is highly extensible and can be seamlessly **adapted to integrate with future advancements**, ensuring long-term relevance and compatibility.
> > >
> > > As for **improvements**, there may be a misunderstanding regarding data pruning, which aims to match the performance of the full dataset without degradation, distinguishing it from other comparisons. In this context, **a 0.3% AP improvement is significant, particularly as performance nears that of the full dataset, where achieving further gains becomes increasingly challenging.** For example, on KITTI, the full dataset achieves 71.5% AP, our method achieves 71.4% AP, and KECOR achieves 71.1% AP. While our method drops only 0.1%, KECOR experiences a 0.4% drop. **Achieving this improvement requires significantly increased labeling efforts, often necessitating an additional 5-10% of labeled data** (our method requires 75%, while other methods may require more than 80%), as shown in Figure 3. This underscores the increasing difficulty of improving performance as it approaches the full dataset's results and validates the efficiency and effectiveness of our approach.
> > >
> > > We hope this clarifies the importance of our contributions and the meaningfulness of the results. Thank you again for your feedback.

---

### Author Response · Authors · 2024-11-25
**Reviewer-Author Discussion Period Ends in TWO Days**

Thanks again for reviewing our paper. We hope that our response adequately addressed your concerns. As the deadline approaches, please let us know if you have any further questions before the reviewer-author discussion period ends. We understand that you are busy, and we would greatly appreciate it if you could consider our response during discussions with the AC and other reviewers.

We are more than happy to address any additional concerns or questions you may have.

Once again, we sincerely thank you for dedicating your time and effort to reviewing our work.

---

### Author Response · Authors · 2024-12-02
**Response Reminder: Approaching Deadline**

**Thank you for reviewing our paper**. As the deadline is approaching, if our response has addressed your concerns, we kindly ask you to consider adjusting your scores accordingly. If you have any further questions or concerns, please let us know, and we will be happy to provide additional clarifications. Your feedback is greatly appreciated, and we look forward to your response before the reviewer-author discussion period ends.

---

### Meta-Review · Area_Chair_G4Xv · 2024-12-15

**Metareview:**

This paper receives 3 negative ratings and 1 positive rating. Although the paper has some merits, e.g., the motivation of the studied topic, the reviewers pointed out a few critical concerns about 1) technical novelty and contributions on uncertainty modeling, 2) experimental results and minor performance gains. After taking a close look at the paper, rebuttal, and discussions, the AC agrees with reviewers' feedback and hence suggests the rejection decision. The authors are encouraged to improve the paper based on the feedback for the next venue.

**Additional Comments On Reviewer Discussion:**

In the rebuttal, some of the concerns like technical clarity are addressed by the authors. However, during the post-rebuttal discussion period, the reviewer uQgi is still not convinced about technical novelty and experiments. Although the other reviewers have not participated actively in the discussions, similar concerns are raised as well. While the authors provided more explanations and results with various settings, the AC agrees with the reviewers that resolving these issues still requires a significant amount of effort for improving the current version to be ready for publication.

---

### Decision · Program_Chairs · 2025-01-22

Reject